# The prevalence of thrombocytopenia and leucopenia among people living with HIV/AIDS in Ethiopia: A systematic review and meta-analysis

**Habtye Bisetegn⦿\*, Hussien Ebrahim⦿**

Department of Medical Laboratory Sciences, College of Medicine and Health Sciences, Wollo University, Dessie, Ethiopia

\* habtiye21@gmail.com

## Abstract

### Introduction

Thrombocytopenia and leucopenia are frequently encountered hematological disorders among people living with HIV/AIDS. This systematic review and meta-analysis were aimed to indicate the national prevalence of thrombocytopenia and leucopenia among HIV/AIDS patients.

### Methods

This systematic review and meta-analysis was conducted following the preferred reporting items for systematic review and meta-analysis (PRISMA) guidelines. A systematic search was conducted from February 01, 2021 to April 02, 2021 using electronic databases Google Scholar, PubMed, Web of Sciences, Google, EMBASE, SCOPUS and ResearchGate. The quality of the included studies was assessed using Newcastle—Ottawa Quality Assessment Scale (NOS) adapted for cross-sectional studies. Data analysis was done using STATA version 14 using metan commands. Random effect meta-analysis was used to estimate the pooled prevalence of thrombocytopenia and leucopenia among people living with HIV/AIDS in Ethiopia.

### Result

Of the 349 initially searched articles, 90 were assessed for eligibility and only 13 articles published from 2014 to 2020 were included in the final meta-analysis. A total of 3854 participants were involved in the included studies. The pooled prevalence of thrombocytopenia was 9.69% (95%CI; 7.40–11.97%). Significant heterogeneity was observed with $I^2$ value of 84.7%. Thrombocytopenia was 11.91% and 5.95% prevalent among HAART naive and HAART exposed HIV/AIDS patients, respectively. The pooled prevalence of leucopenia among HIV/AIDS patients was 17.31% (95%CI: 12.37–22.25%).

**Data Availability Statement:** All relevant data are within the manuscript and its Supporting Information files.

**Funding:** The authors received no specific funding for this work.

**Competing interests:** The authors have declared that no competing interests exist.

**Abbreviations:** AIDS, Acquired Immunodeficiency Syndrome; HAART, Highly Active Anti-Retroviral Therapy; HIV, Human Immunodeficiency Virus.

## Conclusion

This study showed a high prevalence of thrombocytopenia and leucopenia among people living with HIV/AIDS, indicating the necessity of regular screening of HIV seropositive patients for different hematological parameters and providing treatment.

## Introduction

Human Immunodeficiency Virus (HIV) is a retrovirus that attaches itself to CD4 T cell molecules of the immune system and progressively reduces the number of circulating CD4 T helper cells resulting weakened immune system [1]. Human Immunodeficiency Virus (HIV) is characterized by a spectrum of diseases ranging from acute syndrome resulting from the primary infection to a prolonged asymptomatic state and advanced acquired immunodeficiency syndrome (AIDS) [2]. In 2019, approximately 38 million people were living with HIV and ten million died because of the disease and its associated mortality. Out of the 38 million cases, 25.6 million (67%) were in Africa [3].

Hematological abnormalities are widespread clinical and pathological manifestations in people living with HIV/AIDS [4]. As the disease progresses to the advanced stage, it results in disorders of the hematopoietic system and results in thrombocytopenia, leucopenia, anemia, pancytopenia and others. Cytopenia associated with HIV/AIDS can occur due to different mechanisms like a defect in bone marrow production, increased peripheral loss, or destruction of blood cells [5]. Thrombocytopenia is the most frequently encountered hematological abnormality characterized by reduced circulating platelet count leading to bleeding disorders and this can have a direct impact on the patient's mortality, morbidity and quality of life [6]. The severity and prevalence of cytopenia are associated with the stage of the disease and are known to be improved using highly active antiretroviral therapy (HAART) [7]. Thrombocytopenia is a common hematologic disorder in people living with HIV, the mechanism is estimated to be associated with accelerated peripheral platelet destruction and decreased platelet production from impaired megakaryocytes and it can lead to cardiac dysfunction [8, 9]. The disease continued to be highly prevalent in the HAART era, although the severity of the disease decreases with the use of HAART [10]. The prevalence varies with different study settings and ranges from 4.1% to 26.7% [7].

Leucopenia is another common hematological disorder associated with HIV infection with viable prevalence in different study settings [9]. The cause of leucopenia among people living with HIV/AIDS is multifactorial including bone marrow suppression by HIV that leads to decreased granulocyte colony stimulating factor, use of myelosuppressive drugs and opportunistic infections like mycobacterium tuberculosis, leishmaniasis, cytomegalovirus, histoplasmosis, and cryptococcosis [11, 12].

Different studies were conducted to assess thrombocytopenia and leucopenia among people living with HIV/AIDS in Ethiopia with great differences and variable findings. However, there are no study that summarized the pooled prevalence of thrombocytopenia and leukopenia among people living with HIV/AIDS in Ethiopia. This study was aimed to indicate the pooled prevalence of thrombocytopenia and leucopenia among HIV seropositive patients at a national level. Which can alert policy makers and different stack holders for early diagnosis and provision of treatment, which is vital for sustaining the patient's hematopoiesis and alleviating cytopenia.

## Methods

### Search strategy and selection criteria

To identify studies reporting the magnitude of thrombocytopenia and leucopenia among people living with HIV/AIDS, we systematically searched articles from February 01, 2021 to April 02, 2021 by an electronic database search on Google Scholar, PubMed, Web of Sciences, Google, EMBASE, SCOPUS and ResearchGate using the combination of the key terms thrombocytopenia, leukopenia, hematological abnormalities, HIV/AIDS and Ethiopia. Additionally, proceeding of a professional association like the Ethiopian Public Health association (EPHA) and the Ethiopian Medical Laboratory association (EMLA) and national annual conferences were assessed to identify potentially eligible studies. The reference lists of the retrieved articles were also searched for additional articles that might not be addressed by the initial database searches. After completing the initial search, duplicates were removed. The authors independently screened the title and abstract of the searched literatures. Any disagreement was resolved by discussion. The full text of the potentially eligible studies was evaluated based on the inclusion and exclusion criteria. The meta-analysis was carried out according to the preferred reporting items for systematic review and meta-analysis (PRISMA) guidelines [13]. The target population was HIV seropositive Ethiopian people of all ages and the target outcomes were the prevalence of thrombocytopenia and leucopenia.

### Inclusion and exclusion criteria

Original studies reporting the prevalence of thrombocytopenia and leucopenia among people living with HIV/AIDS in Ethiopia were included. We excluded articles published in a language other than English, review articles, non-research letters, and case reports. We also excluded articles if they did not contain vital information that must be extracted and articles whose full text was not available. Studies reporting the prevalence of thrombocytopenia and leukopenia among HIV/AIDS patients co-infected with other diseases such as Malaria, diabetic mellites, hypertension, and *Mycobacterium tuberculosis* were also excluded.

### Data extraction

The two authors independently extracted data by reviewing the full text of potentially eligible studies. If any disagreement occurs, it was resolved through consensus. The following information was extracted using Microsoft Excel: Name of the primary author, year of publication, regions where the study was conducted, study subject (adult, children or all ages), highly active antiretroviral therapy (HAART naïve or on HAART) status, sample size, number of female participants, number of male participants, the prevalence of thrombocytopenia and leucopenia.

### Quality assessment

The two authors independently extracted and reviewed the data of the included studies to ensure consistency. The quality of the included studies was assessed using Newcastle—Ottawa Quality Assessment Scale (NOS) adapted for cross-sectional studies [14]. The quality assessment tool consists of three paraments of quality: "Selection"; that can weight a maximum of five stars, "comparability"; that can weight a maximum of two stars, and "an outcome" that can weight a maximum of three stars. Totally, the tool weights ten stars. The article was graded as "very good quality" if it got nine to ten stars and above, "good quality" if it got seven to eight stars, "satisfactory quality" if it got five to six stars and "unsatisfactory" if it got zero to four

stars. The included studies were very good quality. The quality assessment result is available as an additional file.

## Statistical analysis

After extracting the relevant information in Microsoft Excel 2019, the final meta-analysis was carried out using STATA version 12 with the metan commands. Since significant heterogeneity was noted, the pooled effect size and the corresponding 95% confidence interval (CI) were calculated using a random effect meta-analysis model. The amount of heterogeneity was evaluated using Cochran's Q test and the $I^2$ statistic. The $I^2$ value of less than 25% was equivalent to no heterogeneity, 25 to 50% equivalent to low heterogeneity, 50% to 75% equivalent to moderate heterogeneity and a value greater than 75% was equivalent to high heterogeneity [15, 16]. To explore the source of high heterogeneity observed, subgroup analysis was conducted according to the study subject, region and HAART status. Publication bias was assessed by visual inspection, using the symmetry of the funnel plot, and Egger's test statistics. Asymmetry of the funnel plot and P-value of the Egger's test statistics $\leq 0.05$ indicating the presence of publication bias [17, 18]. Sensitivity analysis was conducted to assess the impact of a single study on the pooled effect size.

## Result

### Identified studies

A total of 349 records were found during the initial search, of which 315 records remained after removing the duplicates. Then 90 records remained after screening by title and abstract of the articles. Of the 90 articles eligible, 77 articles were excluded by the exclusion criteria. A total of 13 articles were included in the final meta-analysis (Fig 1).

### Description of included studies

A total of 13 articles published between 2014 and 2020 were included in this systematic review and meta-analysis. The studies involved 3854 (1643 male and 2211 female) study participants. The sample size of the studies was from 176 to 499. The studies were conducted in three national regional states and one city administration. Of the included studies, nine were conducted in Amhara National regional state, two in Oromia National regional state, one in SNNPR, and one in Addis Ababa city. All the included studies employed a hospital-based cross-sectional study design. Ten studies were performed among adult HIV/AIDS patients while three studies were conducted among children living with HIV/AIDS. Thirteen studies assessed the prevalence of thrombocytopenia among people living with HIV/AIDS. The highest prevalence of thrombocytopenia was reported by Tamir *et al.* [19] from Amhara regional state in 2019 (18.66%) and the lowest was reported by Fenta *et al.* [20] from SNNPR in 2020 (4%). The highest prevalence of leukopenia was reported by Enawgaw *et al.* [12] in 2014 (26.2%) and the lowest was reported by Gebregziabher *et al* [21] in 2017 (4.5%) from Amhara national regional state (Table 1).

### Prevalence of thrombocytopenia in HIV infected patients

The prevalence of thrombocytopenia ranges from 4% (95%CI;1.68–6.32%) to 18.66% (95%CI; 14.85–22.47%). The pooled prevalence of thrombocytopenia among people living with HIV/ AIDS was 9.69% (95%CI; 7.40–11.97%). There was a significant heterogeneity in estimating the pooled effect size with an $I^2$ value of 84.7% and Heterogeneity chi-square of 78.60 (df. = 12) p≤0.000 (Fig 2).

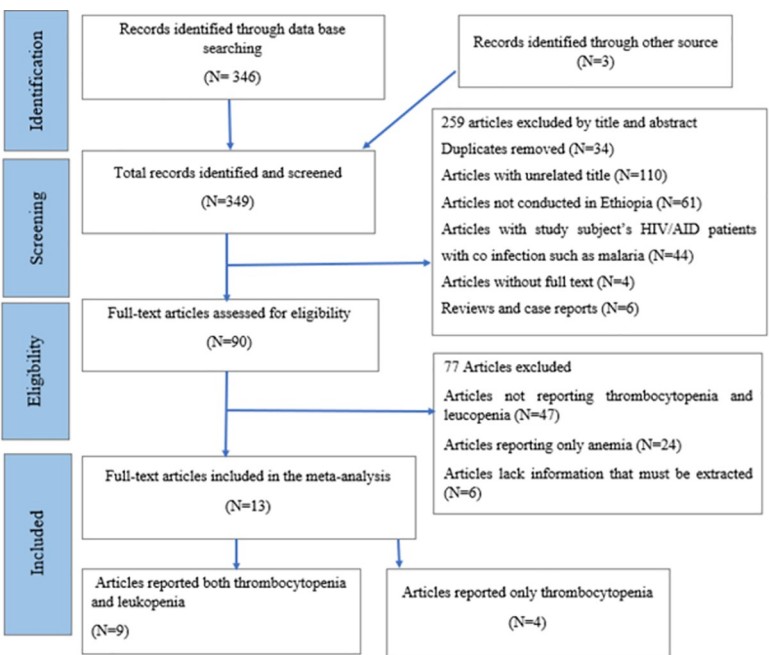

**Fig 1. Flow chart of study selection process.**

## Subgroup analysis

**Thrombocytopenia by HAART status.** In this systematic review and meta-analysis, ten and nine studies assessed the magnitude of thrombocytopenia in HIV/AIDS patients who are taking HAART and who are HAART naïve, respectively. The prevalence of thrombocytopenia among HAART naive participants ranges from 5.9% (95%CI; 3.56–8.24%) to 26% (95% CI;17.4–34.6%) with pooled prevalence of 11.91% (95%CI;8.51–15.31%) (Fig 3). The pooled prevalence was lower in participants taking HAART (5.95%, 95%CI;3.87–8.03%) (Fig 4). In both cases, there were high heterogeneity with $I^2$ of 84.1% and 79.4%, respectively.

**Table 1. Characteristics of the included studies.**

| Authors | Year of publication | Region | Study group | Mean-age (year) | Sample size | HAART status | Thrombocytopenia (%) | Leukopenia (%) |
|---|---|---|---|---|---|---|---|---|
| Gebreweld et al [22] | 2020 | Amhara | Adult | 36 | 499 | On HAART | 12.4 | 13.8 |
| Enawgaw et al [12] | 2014 | Amhara | Adult | 34 | 290 | Both | 6.6 | 26.2 |
| Fenta et al [20] | 2020 | SNNPR | Children | 10.2 | 273 | On HAART | 4 | 14.3 |
| Seyoum et al [23] | 2018 | Amhara | Adult | 35 | 200 | Both | 17 | NA |
| Weyecha et al [24] | 2019 | Oromia | Adult | 38.8 | 308 | Both | 11.4 | 18.2 |
| Fekene et al [25] | 2018 | Oromia | Adult | 34.7 | 361 | Both | 11.1 | 13 |
| Deressa et al [26] | 2018 | Amhara | Adult | 38 | 320 | On HAART | 6.3 | NA |
| Geletaw et al [11] | 2017 | Amhara | Children | 10 | 222 | Both | 9.91 | 24.1 |
| Woldeamanuel et al [27] | 2018 | Addis Ababa | Adult | 40.08 | 176 | On HAART | 5.7 | NA |
| Gebregziabher et al [21] | 2017 | Amhara | Children | 8 | 224 | Both | 8 | 4.5 |
| Wondimeneh et al [28] | 2014 | Amhara | Adult | 33.65 | 390 | HAART naïve | 5.9 | NA |
| Addis et al [29] | 2014 | Amhara | Adult | 33.2 | 189 | HAART naïve | 12.2 | 19 |
| Tamir et al [19] | 2019 | Amhara | Adult | 36.2 | 402 | HAART naïve | 18.66 | 24.4 |

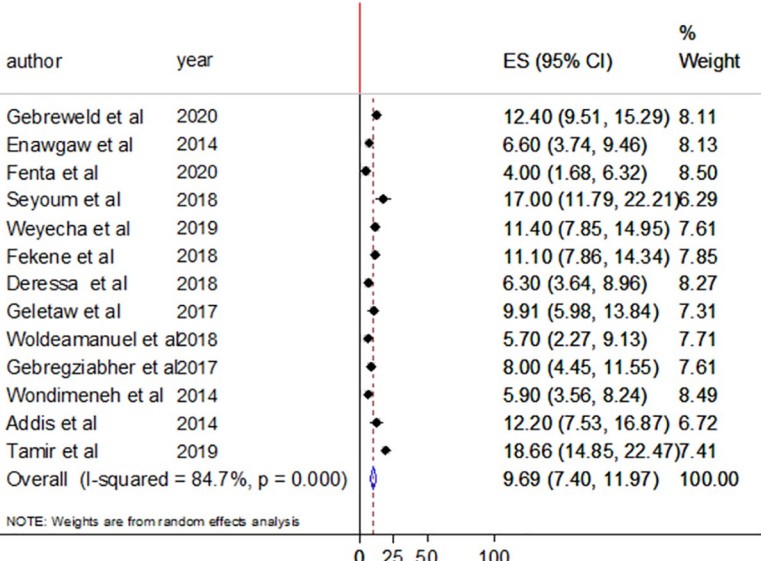

**Fig 2. Forest plot showing the pooled prevalence of thrombocytopenia among people living with HIV/AIDS in Ethiopia.**

Due to high heterogeneity, subgroup analysis was also performed according to region where the study was conducted and the study group (adult or children).

According to subgroup analysis, the pooled prevalence of thrombocytopenia was lower in Amhara region regional state compared to Oromia regional state (10.51% vs 11.24%) (Fig 5). The pooled prevalence of thrombocytopenia was 10.48% in studies involving adult participants and 7.04% in studies conducted among children (Fig 6). The heterogeneity was higher in the studies that involved adult participants and moderate in studies that involved children as participants with $I^2$ values of 85.2% and 74.5% (Fig 6).

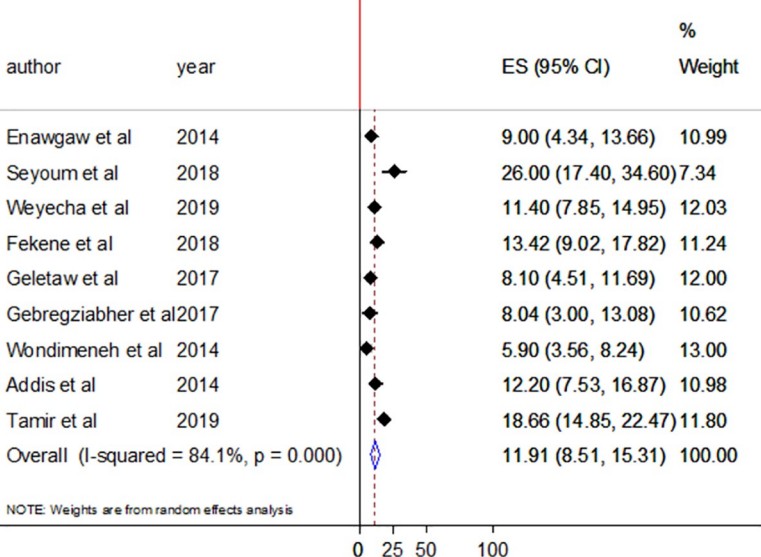

**Fig 3. Forest plot showing the pooled prevalence of thrombocytopenia among HAART naïve HIV/AIDS patients.**

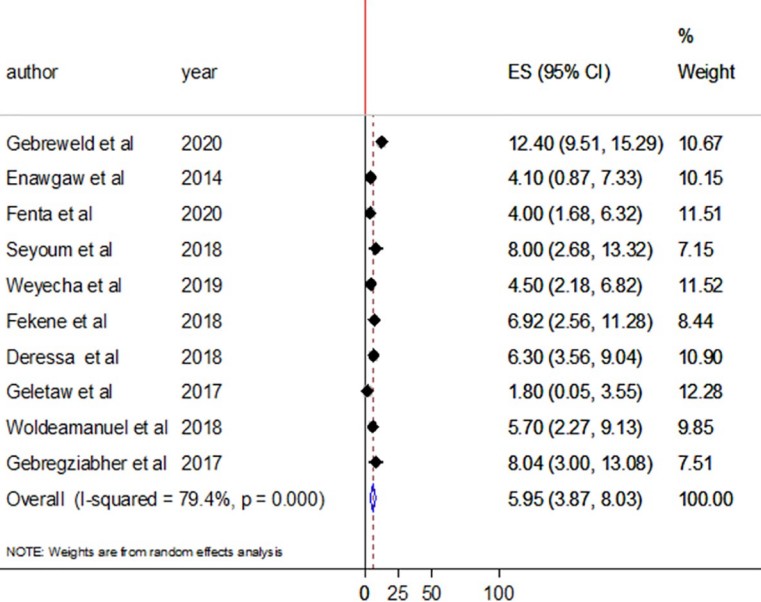

**Fig 4. Forest plot showing the pooled prevalence of thrombocytopenia among HIV/AIDS patients on HAART.**

**Publication bias.** Symmetry of the funnel plot (Fig 7) and Egger's test statistics showed the absence of publication bias with p-value of 0.061.

**Sensitivity analysis.** Sensitivity analysis was done to assess the impact of a single study on the pooled effect size. When each study was omitted independently the resulting pooled effect size was within 95%CI of the combined pooled effect size, this confirms the absence of a single study that affect the pooled effect size (Table 2).

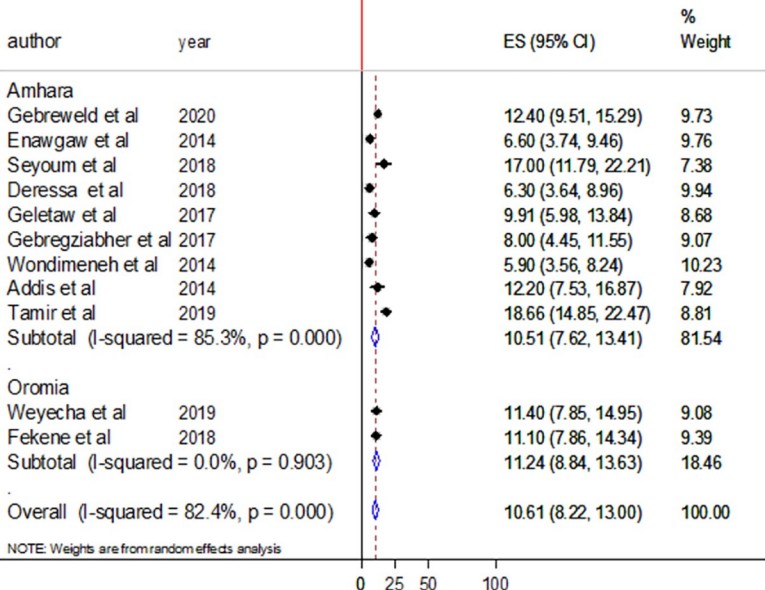

**Fig 5. Forest plot showing subgroup analysis by region.**

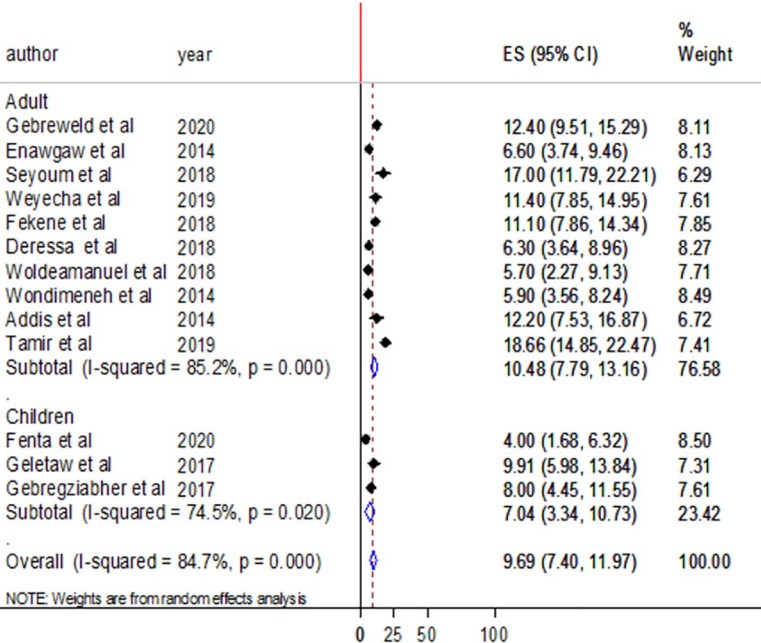

**Fig 6. Forest plot showing subgroup analysis by study group.**

## Prevalence of leucopenia in HIV infected patients

Nine studies assessed the prevalence of leukopenia among people living with HIV/AIDS in Ethiopia. The pooled prevalence of leucopenia was 17.31% (95%CI: 12.37–22.25). There was high heterogeneity with $I^2$ of 92.8% (Fig 8).

## Subgroup analysis

Subgroup analysis was done according to the study group and region where the study was conducted. The pooled prevalence of leucopenia was found to be high among studies involving

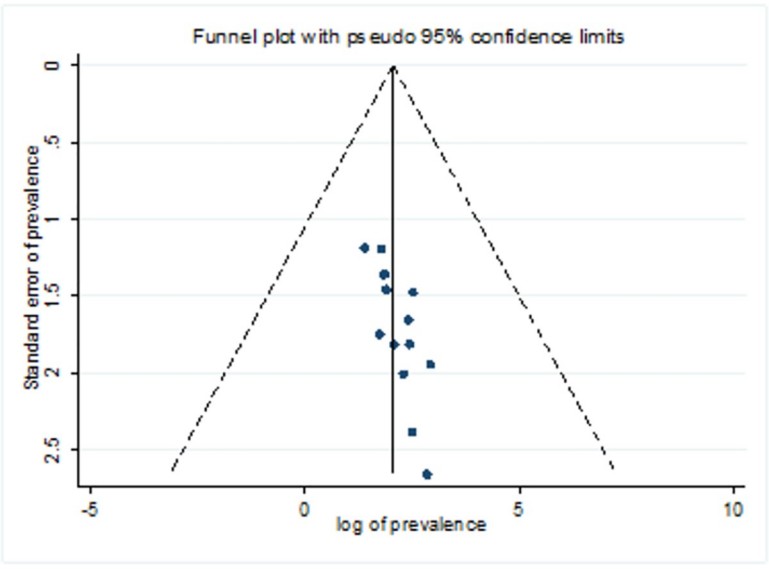

**Fig 7. Funnel plot.**

**Table 2. Sensitivity analysis.**

| Study omitted | Estimate | 95% CI |
|---|---|---|
| Gebreweld et al (2020) | 9.45 | 7.06–11.84 |
| Enawgaw et al (2014) | 9.98 | 7.49–12.46 |
| Fenta et al (2020) | 10.19 | 7.92–12.47 |
| Seyoum et al (2018) | 9.18 | 6.93–11.43 |
| Weyecha et al (2019) | 9.55 | 7.13–11.98 |
| Fekene et al (2018) | 9.58 | 7.13–12.02 |
| Deressa et al (2018) | 10.01 | 7.53–12.49 |
| Geletaw et al (2017) | 9.68 | 7.24–12.13 |
| Woldeamanuel et al (2018) | 10.03 | 7.59–12.46 |
| Gebregziabher et al (2017) | 9.84 | 7.38–12.31 |
| Wondimeneh et al (2014) | 10.05 | 7.58–12.53 |
| Addis et al (2014) | 9.51 | 7.12–11.89 |
| Tamir et al (2019) | 8.87 | 6.9–10.84 |
| Combined | 9.68 | 7.39–11.97 |

adult participants (18.9%) compared to studies that involve children as a participant (14.08%). There was high heterogeneity in both cases with $I^2$ values of 85.4% and 95.5%, respectively (Fig 9). Regarding the region where the studies were conducted, the pooled prevalence of thrombocytopenia was 18.51% and 15.44% in Amhara and Oromia regional states, respectively. Heterogeneity was high in Amhara region with $I^2$ value of 95.4% but it was moderate in Oromia region $I^2$ value of 70.5% (Fig 10).

## Discussion

Hematological abnormalities such as anemia, thrombocytopenia, leucopenia, coagulopathy, neutropenia, and vascular malignancies are encountered in people living with HIV/AIDS [7, 8, 30].

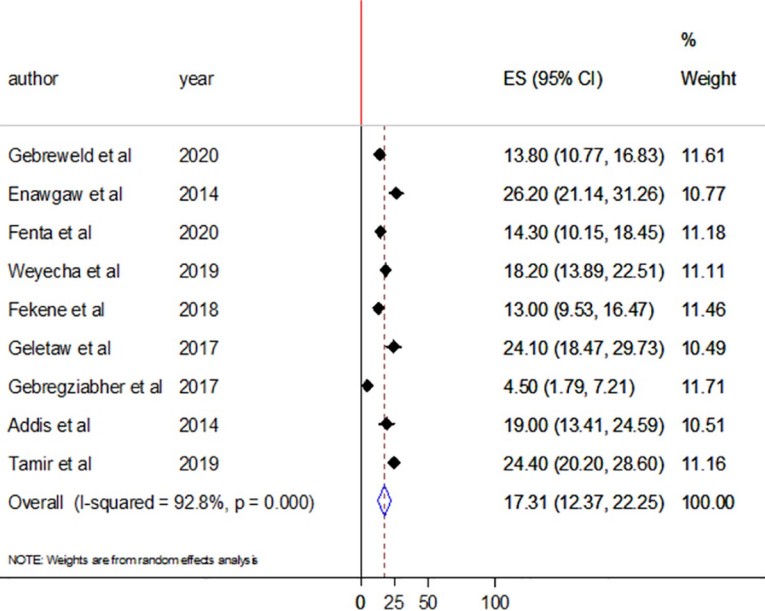

**Fig 8. Forest plot showing the pooled prevalence of leucopenia among HIV/AIDS patients.**

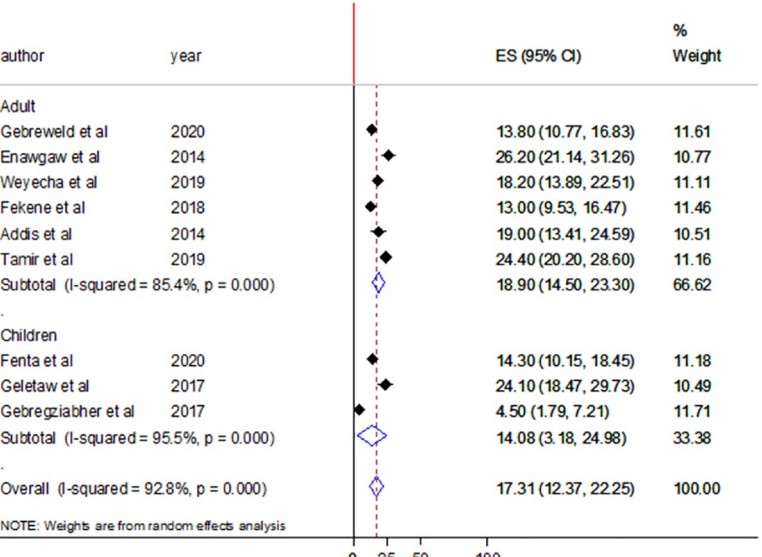

**Fig 9. Forest plot showing the pooled prevalence of leucopenia by study groups.**

Different mechanisms such as the direct intracellular effect of viral infection, increased peripheral destruction of blood cells, interaction with viral proteins at the surface, perturbation of cytokine network, and bone marrow suppression by opportunistic infection are involved in hematological disorders among people living with the virus [9, 31]. In this review, the pooled prevalence of thrombocytopenia was 9.69% (95%CI; 7.40–11.97%). The possible cause of thrombocytopenia could be increased peripheral destruction of platelets, impaired production of platelet from infected megakaryocytes, disturbance of the bone marrow hematopoiesis process, and myelosuppression effect of the medications [8, 9]. This finding was consistent with the estimated range of thrombocytopenia among people living with HIV/AIDS (4.1% to 26.7%) [7]. However,

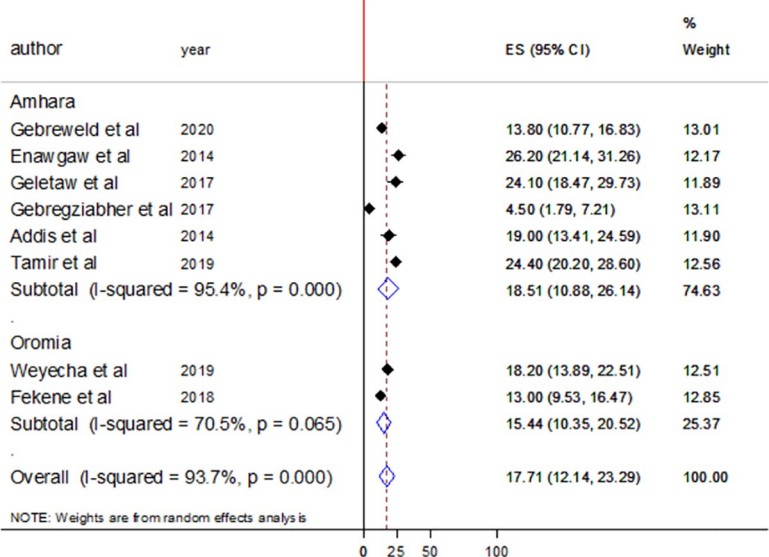

**Fig 10. Forest plot showing the pooled prevalence of leucopenia by study region.**

the finding was lower than the pooled global prevalence of thrombocytopenia among people living with HIV/AIDS ((17.9% (95% CI: 14.69, 21.12)) [32], a report from Tanzania (14.4%) [33] and China (15.6%) [34]. This finding was higher than a study conducted in China that reported 4.5% prevalence of thrombocytopenia in adult HAART naïve participants [35]. The difference might be due to variation in the immune status of the study participants, duration of HAART use, the type of HAART regimens, and variation in the definition of thrombocytopenia.

Subgroup analysis was performed for HAART status, study group, and region of studies. According to subgroup analysis by HAART status, the random effect pooled prevalence of thrombocytopenia was higher in participants who are naïve (11.91%: 95%CI;8.51–15.31%) compared to HAART users (5.95%: 95%CI;3.87–8.03%). The low prevalence of thrombocytopenia among HAART users might be due to the role of HAART in reducing the viral load and improving CD4 T cell count. Highly active antiretroviral therapy is also reported to have a direct role in lowering thrombocytopenia [30]. Although cases are frequently observed, thrombocytopenia typically improved with the use of HAART [36]. In both cases, there was significantly higher between-study variability with $I^2$ of 84.1% and 74.9%, respectively.

In addition, subgroup analysis revealed slightly higher pooled prevalence of thrombocytopenia in Oromia regional state than in Amhara regional state (11.24% vs 10.51%). The difference might be due to the variation number of studies; in Oromia region only two studies were conducted while in Amhara region there were 9 studies.

The results of subgroup analysis also revealed a high pooled prevalence of thrombocytopenia among studies involving adult participants than studies that involve children as a participant (10.48% vs7.04%). The difference might be due to the variation in immune status, that adults have a well-developed immune system compared to the immune system of children.

Symmetry of the funnel plot and the Egger's test statistics concluded the absence of publication bias in this study. Sensitivity analysis was also performed and indicated the absence of a single study impact on the overall pooled effect size.

In this systematic review and meta-analysis, nine studies reported leucopenia among HIV/AIDS patients. Random effect meta-analysis showed 17.31% (95%CI: 12.37–22.25) pooled prevalence of leucopenia among people living with HIV in Ethiopia. Between study heterogeneity was higher ($I^2$ = 92.8%). The mechanism of leukopenia in HIV/AIDS patients can be inhibition of hematopoietic progenitor colony formation and differentiation, alteration of the stromal microenvironment that supports hematopoiesis, decreased cellularity, myelodysplasia and inhibition of production/expression of factors that stimulate white blood cell production [9]. This finding was lower than reports from Tanzania (23.6%) [33] and China (33.2%) [34]. The observed difference might be due to the difference in immune status of the study participants, difference in nature of the study participant, and the clinical stage of AIDS. Leucopenia is known to be associated with the advanced stages of AIDS [1]. The pooled prevalence was also indicated by subgroup analysis with regions and study groups. Of the nine studies that reported leucopenia, six assessed leucopenia among adult and three among children. Regarding region, six studies were conducted in Amhara regional state, two in Oromia regional state, and one in SNNPR. The pooled prevalence of leucopenia was 18.9% among adult and 14.08% among children. A 18.51% pooled prevalence of leucopenia was found from studies conducted in Amhara regional state.

This study was done with the following limitations. Initially, there was high heterogeneity even after subgroup analysis for some variables. Then, it was not able to assess factors associating with the pooled prevalence of thrombocytopenia and leucopenia in HAART user participants such as clinal stage of AIDS, CD4 T cell count, type of HAART regimen, and duration of HAART regimen. Finally, only studies published in English language were included this may expose the study to language bias.

## Conclusion

This systematic review and meta-analysis showed a high prevalence of thrombocytopenia and leukopenia among people living with HIV/AIDS in Ethiopia. This is a clear message that regular screening of HIV seropositive patients for different hematological parameters and providing treatment and subsequent follow-up is crucial for reducing HIV associated mortality and to improve quality of life for an individual living with virus. In addition, our finding indicated the importance of early diagnosis of HIV and timely initiation of HAART to reduce the advancement of HIV disease and its subsequent hematological complications.

## Supporting information

**S1 Checklist. PRISMA checklist.**
(DOC)

**S1 File. Quality assessment result of included studies.**
(DOCX)

## Acknowledgments

The authors would like to forward their heartful gratitude to the authors of the original research as well as their study participants. We would like to thank all who helped us in completing this systematic review and meta-analysis.

## Author Contributions

**Conceptualization:** Habtye Bisetegn.

**Data curation:** Habtye Bisetegn.

**Formal analysis:** Habtye Bisetegn, Hussien Ebrahim.

**Funding acquisition:** Habtye Bisetegn.

**Investigation:** Habtye Bisetegn.

**Methodology:** Habtye Bisetegn.

**Software:** Habtye Bisetegn, Hussien Ebrahim.

**Supervision:** Habtye Bisetegn.

**Validation:** Habtye Bisetegn, Hussien Ebrahim.

**Visualization:** Habtye Bisetegn.

**Writing – original draft:** Habtye Bisetegn.

**Writing – review & editing:** Hussien Ebrahim.

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
