## [Decision Letter · Decision Letter 0]

13 Aug 2021

PONE-D-21-13608

The Prevalence of Thrombocytopenia and Leucopenia among People living with HIV/AIDS in Ethiopia: A Systematic Review and Meta-analysis

PLOS ONE

Dear Dr. Bisetegn,

Thank you for submitting your manuscript to PLOS ONE. After careful consideration, we feel that it has merit but does not fully meet PLOS ONE’s publication criteria as it currently stands. Therefore, we invite you to submit a revised version of the manuscript that addresses the points raised during the review process.

We look forward to receiving your revised manuscript.

Kind regards,

Ahmet Emre Eşkazan, M.D.

Academic Editor

PLOS ONE

Journal Requirements:

2. In your Methods section, please provide additional information on the beginning and end dates for inclusion of articles, and the justification for why these were chosen.

5. We suggest you thoroughly copyedit your manuscript for language usage, spelling, and grammar. If you do not know anyone who can help you do this, you may wish to consider employing a professional scientific editing service. Whilst you may use any professional scientific editing service of your choice, PLOS has partnered with both American Journal Experts (AJE) and Editage to provide discounted services to PLOS authors. Both organizations have experience helping authors meet PLOS guidelines and can provide language editing, translation, manuscript formatting, and figure formatting to ensure your manuscript meets our submission guidelines. To take advantage of our partnership with AJE, visit the AJE website (http://learn.aje.com/plos/) for a 15% discount off AJE services. To take advantage of our partnership with Editage, visit the Editage website (www.editage.com) and enter referral code PLOSEDIT for a 15% discount off Editage services. If the PLOS editorial team finds any language issues in text that either AJE or Editage has edited, the service provider will re-edit the text for free. Upon resubmission, please provide the following: The name of the colleague or the details of the professional service that edited your manuscript A copy of your manuscript showing your changes by either highlighting them or using track changes (uploaded as a *supporting information* file) A clean copy of the edited manuscript (uploaded as the new *manuscript* file)

Reviewers' comments:

Reviewer's Responses to Questions

**Comments to the Author**

1. Is the manuscript technically sound, and do the data support the conclusions?

Reviewer #1: Yes

Reviewer #2: Yes

2. Has the statistical analysis been performed appropriately and rigorously? 

Reviewer #1: I Don't Know

Reviewer #2: Yes

3. Have the authors made all data underlying the findings in their manuscript fully available?

Reviewer #1: Yes

Reviewer #2: Yes

4. Is the manuscript presented in an intelligible fashion and written in standard English?

Reviewer #1: Yes

Reviewer #2: Yes

5. Review Comments to the Author

Reviewer #1: The authors have prepared a well written manuscript conducting a rigorous meta analysis of leucopenia and thrombocytopenia in Ethiopia among PLWH

Major comments

1) widen the axis of the forest plots so the figures can better how differences between the studies

2) Individuals of African descent have lower normal levels of white blood cells. IT would be important to note how many studies used locally determined norms in determining rates of leucopenia.

3) A discussion of reports of leucopenia/thrombocytopenia prevalence in healthy people living without HIV in Ethiopia and/or Africa would improve the quality of the manuscript.

4) The findings of lower prevalence rates of hematological abnormalities associated with ART are important and are supportive of early diagnosis and ART but it is probably inappropriate to state that your findings "prove" that ART causes fewer hematological abnormalities or that ART is important

Minor comments

1) Under Identified studies: can delete "were" from "were remained" in the first two sentences of that section

2) Should Be "description of included studies" instead of Discerption.

3) In that section, the sentence "Threaten studies..." is confusing and should be reworded

4) Figure 1. 76 Articles excluded, not 76 Article excluded

5) Capitalize China throughout

Reviewer #2: 1. Introduction: Write the information about case and effect of Leucopenia disorder, as written for Thrombocytopenia ( is a common hematologic disorder in people living with HIV, the mechanism is estimated to be associated with accelerated peripheral platelet destruction and decreased platelet production from impaired megakaryocytes and it can

lead to cardiac dysfunction)

2. Result: Articles excluded were mentioned 77 in result where as in Figure-1 it is 76, also the break up are not matching.

3. Discerption of included studies: The highest prevalence of leukopenia reported was 26.2% not 24.4%, also change the reference citation.

6. PLOS authors have the option to publish the peer review history of their article (what does this mean?). If published, this will include your full peer review and any attached files.

Reviewer #1: **Yes: **Julie A. Ake

Reviewer #2: No

---

## [Author Response · Author response to Decision Letter 0]

19 Aug 2021

Point by point response to reviewers and editors 

Dear Editor and reviewers

Frist we would like to give our deepest gratitude to the reviewers and editors for your valuable and constructive comments. 

We have gone through all your comments and correct the manuscript as per your comment. We have worked hard on the language of the manuscript by consulting a person who is proficient in English.

 Note: In the revised manuscript we showed the recent changes we made by making the font color red and sheading it with yellow color. 

Editor Comment

Thank you very much for your positive feedback to our work. We have carried out the minor revision based on your and reviewer’s comments. Thank you again.

In the method section we have indicated the beginning and end date of the literature search. There was no restriction in time for studies to be included in the meta-analysis.

Reviewer #1

All the issue you raised are very important for the improvement of the manuscript and we have accepted all and wrote the manuscript accordingly.

I have increased the length of all the graphs. 

All the included studies used the world health organization (WHO) cut off values in determining leucopenia. Yes, you are right it was very good if the authors use the locally determined normal values to determine the rate of leucopenia. 

I have tried to find articles that assessed thrombocytopenia and leucopenia but I didn’t got articles that reported thrombocytopenia and leucopenia in healthy people living without HIV/AIDS.

The excluded articles were 77. It was incorrect in the graph, thank you for your direction we have corrected it. 

The spelling, punctuation and other language errors are corrected accordingly. 

Reviewer #2

All the issue you raised are very important for the improvement of the manuscript and we have accepted all and wrote the manuscript accordingly.

In the introduction we have added the possible cause of leucopenia among people living HIV/AIDS.

You are correct the highest prevalence is 26.2%, sorry for the error and we have corrected it.

The spelling, punctuation and other language errors are corrected accordingly.

The excluded articles were 77. It was incorrect in the graph, thank you for your direction we have corrected it.

---

## [Decision Letter · Decision Letter 1]

6 Sep 2021

The Prevalence of Thrombocytopenia and Leucopenia among People living with HIV/AIDS in Ethiopia: A Systematic Review and Meta-analysis

PONE-D-21-13608R1

Dear Dr. Bisetegn,

We’re pleased to inform you that your manuscript has been judged scientifically suitable for publication and will be formally accepted for publication once it meets all outstanding technical requirements.

Kind regards,

Ahmet Emre Eşkazan, M.D.

Academic Editor

PLOS ONE

Additional Editor Comments (optional):

Reviewers' comments:

Reviewer's Responses to Questions

**Comments to the Author**

1. If the authors have adequately addressed your comments raised in a previous round of review and you feel that this manuscript is now acceptable for publication, you may indicate that here to bypass the “Comments to the Author” section, enter your conflict of interest statement in the “Confidential to Editor” section, and submit your "Accept" recommendation.

Reviewer #1: All comments have been addressed

Reviewer #2: All comments have been addressed

2. Is the manuscript technically sound, and do the data support the conclusions?

Reviewer #1: Yes

Reviewer #2: Yes

3. Has the statistical analysis been performed appropriately and rigorously? 

Reviewer #1: Yes

Reviewer #2: Yes

4. Have the authors made all data underlying the findings in their manuscript fully available?

Reviewer #1: Yes

Reviewer #2: Yes

5. Is the manuscript presented in an intelligible fashion and written in standard English?

Reviewer #1: Yes

Reviewer #2: Yes

6. Review Comments to the Author

Reviewer #1: (No Response)

Reviewer #2: All comments are addressed I am fully satisfied with authors reply. It may accepted for publication.

7. PLOS authors have the option to publish the peer review history of their article (what does this mean?). If published, this will include your full peer review and any attached files.

Reviewer #1: No

Reviewer #2: No

---

## [Editor Report · Acceptance letter]

10 Sep 2021

PONE-D-21-13608R1 

The Prevalence of Thrombocytopenia and Leucopenia among People living with HIV/AIDS in Ethiopia: A Systematic Review and Meta-analysis 

Dear Dr. Bisetegn:

I'm pleased to inform you that your manuscript has been deemed suitable for publication in PLOS ONE. Congratulations! Your manuscript is now with our production department. 

Kind regards, 

on behalf of

Dr. Ahmet Emre Eşkazan 

Academic Editor

PLOS ONE